# Factors associated with knowledge and practices of COVID-19 prevention among mothers of under-2 children in Bangladesh

**Zarin Tasnim** *⊕, **Muhammed Nazmul Islam**⊕, **Antara Roy, Malabika Sarker**

BRAC James P Grant School of Public Health, BRAC University, Dhaka, Bangladesh

⊕ These authors contributed equally to this work.
* zarintasnim45@gmail.com

## Abstract

The COVID-19 pandemic extensively impacted maternal, neonatal, and child health (MNCH) in Bangladesh. Misconceptions arising from a lack of knowledge related to the virus contributed to reduced uptake of MNCH services, which eventually helped increase maternal and neonatal mortality rates during the pandemic. In this study, we assessed the knowledge and practices related to COVID-19 prevention among the mothers of under-2 children in Bangladesh. The study was conducted in May 2021 as part of a broader research project related to COVID-19 response on MNCH service utilization. We collected data from 2207 mothers in six districts of Bangladesh using a multi-stage cluster sampling technique. We constructed weighted and unweighted composite knowledge and practice scores and identified different socio-demographic characteristics associated with the scores using multilevel generalized mixed-effect linear regression models. In general, the mothers revealed poor knowledge and practices related to COVID-19. On a weighted scale of 100, the mean composite knowledge and practice scores were 32.6 (SD = 16.4) and 53.1 (SD = 13.9), respectively. The mothers presented inadequate knowledge about COVID-19 transmission, symptoms, and the recommended preventive measures. At the same time, maintaining a safe physical distance was the least practiced preventative measure (10.3%). Level of education, access to television, and the internet were significantly positively associated with their knowledge and practices related to COVID-19. Knowledge score was also positively associated with the practice score (OR = 1.26; *p*-value <0.001). Mothers living in islands or wetlands scored poorly compared to those living in inland. The results indicate significant gaps in knowledge and practices related to COVID-19 prevention among mothers of under-2 children. Addressing these gaps, particularly by targeting mothers with lower levels of education and residing in hard-to-reach geographic locations, could consequently help enhance MNCH service uptake during pandemics like COVID-19.

## Introduction

The Coronavirus Disease 2019 (COVID-19) pandemic has altered the course of our history by affecting more than 767 million people and claiming about 7 million lives globally [1]. Around

**Data Availability Statement:** All relevant data are within the paper and its Supporting information files.

**Funding:** The study was supported by the Foreign, Commonwealth, & Development Office (FCDO), UK. BRAC James P Grant School of Public Health (JPGSPH), BRAC University was a sub-grantee of this fund and received the fund from BRAC's Health, Nutrition, and Population Program (HNPP) to conduct the study (grant number: 205268-113, PO 40125468). The funders had no role in study design, data collection and analysis, decision to publish, or preparation of the manuscript.

**Competing interests:** The authors have declared that no competing interests exist.

28% of the global cases and 22% of the overall deaths were from Asia [2]. In particular, Bangladesh accounted for more than 2 million positive cases and about 30 thousand deaths [2].

In addition to these immediate consequences, in Bangladesh, the COVID-19 pandemic was responsible for a drastic decline in the uptake of essential healthcare services, notably in the field of maternal, neonatal, and child health (MNCH) [3–5]. Antenatal care visits dropped by 67%, breastfeeding counseling decreased by 53%, and immunization uptake decreased by 66.1% [6]. Moreover, institutional deliveries and postnatal care visits also significantly declined during the pandemic [4]. These collective declines eventually increased maternal and child mortality and morbidity in Bangladesh [3,4,7].

The conditions were similar in many lower- and lower-middle-income countries since their health literacy was low [8]. The utilization of MNCH services was further worsened by misinformation that emerged on social media and spread faster than scientifically valid information [9]. During a health emergency such as the COVID-19 pandemic, women living in such resource-poor settings are considered comparatively more vulnerable and marginalized due to their limited access to information [10–12].

To curtail COVID-19-related morbidity and mortality, the World Health Organization (WHO) recommended some specific non-pharmacological interventions along with vaccination [13]. These non-pharmacological approaches included proper mask-wearing, frequent and appropriate hand-washing with an alcohol-based solution or soap, and maintaining a safe physical distance [14]. However, appropriate adoption of these non-therapeutic behavioral interventions is greatly influenced by people's knowledge, attitude, and practice [15–18].

Several studies have been conducted on COVID-19 preventive knowledge, attitude, and practices in various target populations, such as adults, students, healthcare workers, women, and slum dwellers [19–24]. However, to ensure proper utilization of MNCH services, mothers of young children must have knowledge of and adherence to the specific COVID-19 preventive measures, which ultimately is influenced by their knowledge and practices [25,26].

In this study, we assess the COVID-19 preventive knowledge and practices among the mothers of under-2 children in Bangladesh and identify the factors associated with their knowledge and practices. The findings could help develop interventions targeted towards mothers, neonates, and children, improving knowledge and practices in preventing COVID-19. Eliminating misconceptions through proper knowledge and practices could help enhance MNCH service utilization in Bangladesh and similar countries during pandemics like COVID-19.

## Methods

### Study sites, population, and sampling

During the pandemic in 2020, six districts of Bangladesh, namely Narayanganj, Kishoreganj, Sherpur, Bogura, Bagerhat, and Bhola, had high COVID-19 infection rates and low MNCH service utilization rates. BRAC initiated a community-based COVID-19 response program in these districts to improve their COVID-19 preventive practices and MNCH service utilization (Fig 1). A research was commissioned to assess the effects of this program on MNCH service utilization using a repeated cross-sectional study design.

Among different MNCH services, the postnatal care (PNC) service utilization within two days after delivery was particularly low in Bangladesh (i.e., 52% according to Bangladesh Demographic and Health Survey, 2017–18) [28]. Hence, for that research, the sample size was calculated using PNC service utilization as the outcome variable. The primary target population was the mothers of children aged 2 years or less. For each wave of data, the targeted sample size was 2400 households from six districts (i.e., 400 households per district). In this paper,

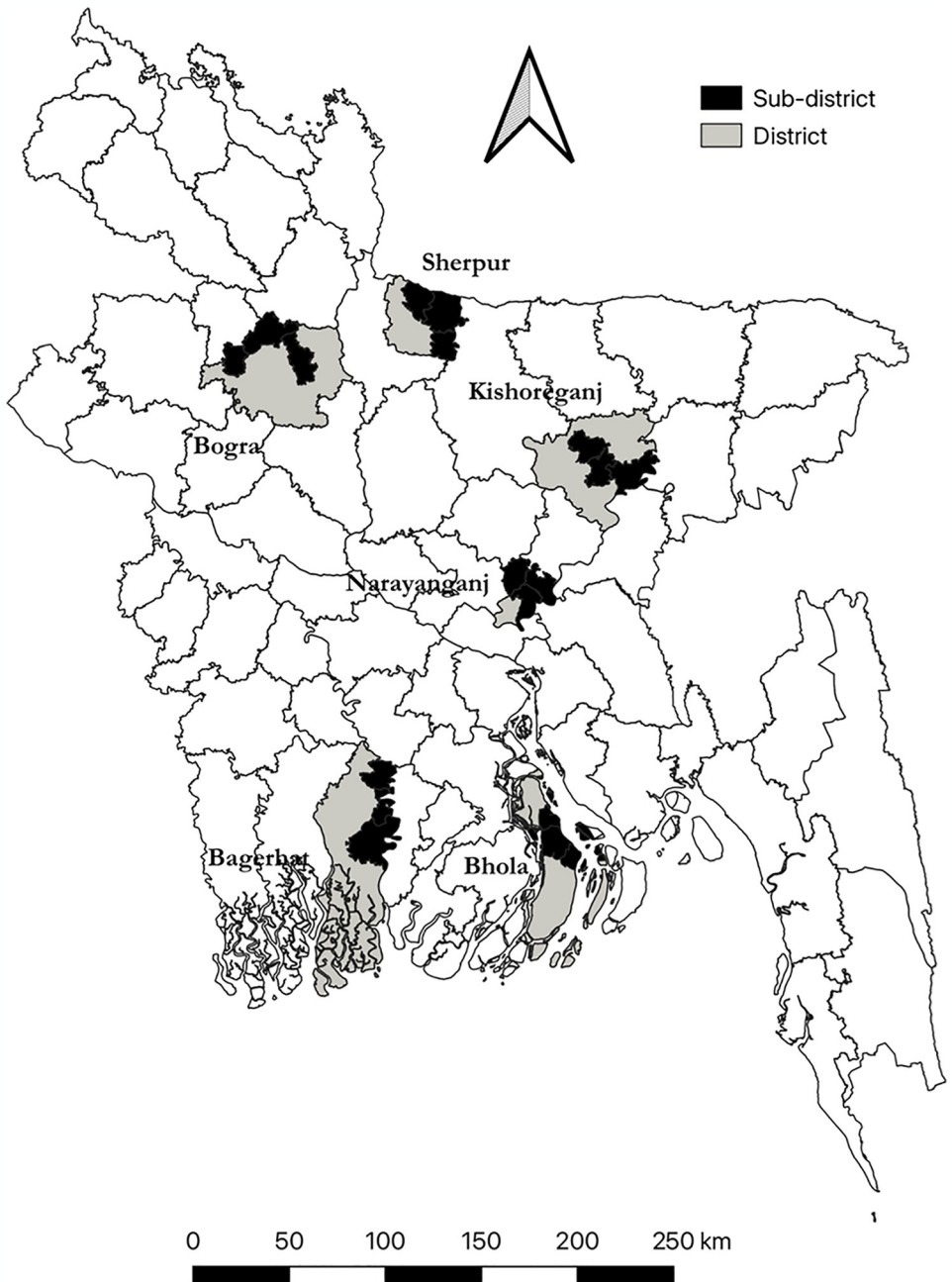

**Fig 1. Study sites. Note:** (a) The map of study sites was created with QGIS using the public domain basemap shapefile published by the Humanitarian Data Exchange (HDX) [27].

we used the wave collected between 9th May and 28th May 2021 as a follow-up survey for the program evaluation initiative mentioned above. We used the data to assess the knowledge and practices related to COVID-19 among the mothers of under-2 children. This paper succeeded a similar assessment conducted using the baseline data, which provided all the details on sample size calculation [29].

The sampled households were selected using a multi-stage cluster sampling technique where each village was considered a cluster. We selected 25 households from each of the 96

villages to survey 2400 households. First, we randomly selected three sub-districts from each of the six districts. Then, again, we randomly selected four unions from each sub-district. Using the population-proportion-to-size (PPS) technique, we calculated the required number of villages (i.e., clusters) for each union, where the total number of households residing in a union was used to represent size. Then, we randomly selected the villages for the unions. Finally, 25 households that had a mother of an under-2 child were chosen from a village following a systematic random sampling approach. We used BRAC program's household lists as a sampling frame to select the households. A team of 60 trained surveyors collected the data using SurveyCTO version 2.70. To cross-check the data, field supervisors revisited 5% of households during the data collection. After excluding observations of four incomplete surveys, we finally used data from 2207 mothers of under-2 children.

## Variables and their measurements

To assess knowledge and practices related to COVID-19 prevention, we prepared a questionnaire based on internationally recommended guidelines for COVID-19 [14]. We asked eight questions to assess a participant's knowledge related to the transmission, symptoms, management, and prevention of COVID-19. The questions were both single-select and multiple-select in nature and had 49 correct options in total (see S1 Table). We assessed a participant's handwashing, mask-wearing, and physical distancing practices using 10 questions containing 20 correct options (see S2 Table). We counted the correct responses to construct unweighted knowledge and practice scores.

However, since the number of questions as well as their correct responses varied for different domains of knowledge and practice, we balanced the scores by weighing the domains equally. The weighted knowledge and practice scores can be compared on a scale of 100. First, the total weight for knowledge was equally split between its two domains: transmission, symptoms, and management (50%) and prevention of COVID-19 (50%). In contrast, the total weight for practice was equally distributed among its three domains: hand washing (33.33%), mask-wearing (33.33%), and physical distancing (33.33%). Then, the weight assigned for a particular domain was again equally distributed among its sub-domains. For example, the domain transmission, symptoms, and management was constituted of three equally weighted subdomains: transmission (16.667%), symptoms (16.667%), and management (16.667%). Finally, the weight assigned for a sub-domain was equally distributed among the questions and/or responses that belonged to it. The unweighted scores of the questions were multiplied by their assigned weights to generate the weighted scores; aggregating them resulted in a weighted composite score of 100 (for details, see S1 and S2 Tables).

Besides knowledge and practice, we asked questions about different demographic and socioeconomic features such as age, education, occupation, household size, and monthly household income. We also enquired about the education and occupation of the household head. To check whether the participant had access to information, we asked about their access to television, phone, or internet connection. Moreover, we asked whether their household shares the kitchen and toilet with other households.

## Statistical analysis

First, we conducted a descriptive analysis of all the demographic- and socioeconomic features and specific questions on knowledge and practices related to COVID-19. Then, we descriptively analyzed the weighted and unweighted scores. We reported the mean scores and standard deviations for both aggregated knowledge and practice scores and their sub-domains. We

plotted histograms of the distribution of scores and compared them against the standard normal distribution.

Finally, we identified different background characteristics associated with the scores using multilevel generalized linear mixed-effects models. We asked eight knowledge questions with 49 correct options and 10 practice questions with 20 correct options. The number of correct responses was assumed to follow a Bernoulli distribution for each question. We modeled the odds of correct responses using a logit link function. The data structure was hierarchical, representing three levels—responses to different questions were nested in individuals and individuals were nested in villages. The models took the following functional form:

$$g\left\{E\left(Y_{ijk}|X_{ijk}, c_{jk}, d_k\right)\right\} = \beta_0 + \boldsymbol{\beta_x}\boldsymbol{X_{ijk}} + c_{jk} + d_k + \boldsymbol{\varepsilon}_{ijk}$$

where $Y_{ij}$ is the dependent variable (i.e., binary response for a knowledge or practice option) for a question $i$ for an individual $j$ residing in village $k$; $\boldsymbol{X_{ij}}$ represents the corresponding vector of observable background features such as age, education, occupation, household size, monthly household income, etc.; $c_j$ and $d_k$ are the random intercepts for individual- and village-level, respectively; and $\varepsilon_{ij}$ is the random disturbance. The models used a link function, $g\{.\}$ which was assumed to follow a Bernoulli distribution. All the random intercepts were assumed to be normally distributed. We reported the adjusted Odds ratios with corresponding P-values and 95% confidence intervals. All the analyses were carried out in Stata version 17.

## Ethical considerations

The ethical review committee of the BRAC James P Grant School of Public Health (JPGSPH) at BRAC University, Bangladesh, approved the study protocol. The reference number is IRB-20-November'20-049. Before starting any data collection activity, authorizations were collected from the respective local administrative bodies of the selected sub-districts. Informed written consent was also obtained from the participants of this study. All procedures performed in this study involving human participants were in accordance with the ethical standards of JPGSPH, BRAC University, and with the 1964 Helsinki Declaration and its later amendments or comparable ethical standards. Moreover, since the survey was conducted during the COVID-19 pandemic, all the recommended protective measures (e.g., wearing face masks and maintaining a physical distance of at least 2 meters) were taken while interviewing the participants.

## Results

### Sociodemographic characteristics

Table 1 demonstrates the sociodemographic profile of the respondents. The mean age of the mothers was approximately 25 years (SD 5.5, Range 16–46 years). A big chunk of the respondents were housewives (94.15%). More than half of the mothers accomplished secondary education (52.4%), however, 7.7% of them did not receive any formal education. On the other hand, more than one-third (36.8%) self-reported household heads acquired primary education; only 8% obtained higher secondary (HSC) education or above. A large share of the household heads were wage workers (37.2%) and engaged in agriculture (26.4%). The average monthly household income was 15504 BDT (SD 17060). Almost all of them owned mobile phones (98.7%) and approximately half had access to television and the internet.

### COVID-19 preventive knowledge

For COVID-19 transmission, although nearly half of the respondents (48.98%) knew about both direct and indirect transmission, 16.18% of them had no knowledge of either. More than

**Table 1. Sociodemographic characteristics.**

|  | Mother |
|---|---|
|  | **N = 2207** |
| Age [years; Mean (SD)] | 25.2 (5.5) |
| **Education** |  |
| No formal education | 7.7 |
| Primary or less | 28.7 |
| Above primary up to SSC | 52.4 |
| HSC and above | 11.2 |
| **Occupation** |  |
| Home-maker | 94.9 |
| Work outside | 5.1 |
| **Head's education** |  |
| No formal education | 26.2 |
| Primary or less | 36.8 |
| Above primary up to SSC | 29.0 |
| HSC and above | 8.0 |
| **Head's occupation** |  |
| Wage-worker | 37.2 |
| Agriculture | 26.4 |
| Service-holder | 12.6 |
| Self-employed | 18.5 |
| Home-maker | 1.6 |
| Others | 3.7 |
| Household size [members, Mean (SD)] | 5.3 (1.8) |
| Household income [BDT., Mean (SD)] | 15,504 (17,060) |
| = 1 if has access to television | 50.3 |
| = 1 if has access to mobile phone | 98.7 |
| = 1 if has access to internet | 50.8 |
| = 1 if shares kitchen with others | 18.5 |
| = 1 if shares toilet with others | 35.8 |
| **District** |  |
| Bhola | 12.4 |
| Narayanganj | 13.8 |
| Bagerhat | 12.7 |
| Bogura | 29.5 |
| Kishoreganj | 21.3 |
| Sherpur | 10.3 |

**Note:** (a) Unless specified otherwise, all the values are reported in percentages; (b) standard deviation of the scores are reported in parentheses.

one-fifth of the mothers (21.84%) thought the individual affected by COVID-19 would always reveal symptoms. The most frequently reported COVID-19 symptoms by the respondents were fever or chills (87.86%), cough (80.79%), sore throat (47.76%), shortness of breath (44.99%), and headache (25.69%). A notable proportion of the respondents indicated staying at home (54.96%) and consulting a health care provider (42.86%) as the home remedy options for the symptoms that mimicked COVID-19. As preventative measures against COVID-19, nearly two-thirds of respondents indicated regular hand-washing with an appropriate solution

(62.26%), and half of them indicated wearing a face mask outside the home (55.55%) and maintaining distance from others (46.67%). Additionally, the mothers were questioned regarding the proper techniques for wearing a mask, washing hands, and maintaining physical distance (S1 Table).

Approximately, 43% of the participants were not aware of covering their nose, mouth, and chin while wearing the mask. In addition, a vast majority of them (82.42%) did not have any idea regarding not sharing their masks with other people. Overall, 84.28% mothers had knowledge of using enough soap while washing hands and only 29.95% recalled the information to scrub hands for at least 20 seconds. Furthermore, more than one-third of them were aware of staying at minimum one-meter distance from others (39.74%) to prevent COVID-19. However, 17.58% did not have any idea about proper mask-wearing techniques, 7.84% on hand-washing techniques, and 39.74% on the concept of physical distancing. The correct responses to the survey questions on COVID-19 knowledge have been reported in S2 Table.

### COVID-19 preventive practice

Almost all the participants had soap (99.37%) at their home. However, most of them did not clean their hands before touching eyes, nose, and mouth (90.35%). Although majority of the participants had masks at their homes (86.27%), a large number of them shared their masks with their family members (97.42%). Around 30% respondents always used a face mask while going out of their house, nonetheless, 28.55% never used any face mask. On the other hand, more than one-third of the mothers joined social gatherings (39.87%) in the past two weeks and 48.62% did not keep a safe physical distance from others. The correct responses to the survey questions on COVID-19 preventive practices have been illustrated in S2 Table.

### Knowledge and practice scores

The distribution of weighted and unweighted knowledge and practice scores has been illustrated in Figs 2–5. The overall mean weighted knowledge score was 32.6 (SD 16.4) on a scale of 100. The distribution of standardized knowledge scores was found to be approximately normal (Figs 2 and 3). Almost 65.93% mothers scored within the range of -1 SD and +1 SD (between the score of 16.2 and 49) and 97.28% within -2 SD to +2 SD on the COVID-19 preventive knowledge. The findings also indicated poor knowledge of COVID-19 transmission, symptoms, and management (mean score 16.4, SD 8.3) and COVID-19 prevention (mean score 16.2, SD 10.1) (Table 2).

On the contrary, the practice score was defined by three domains: mask-wearing, hand-washing, and physical distancing. The overall mean weighted practice score of the mothers was 53.1 (SD 13.9) on a scale of 100 (Table 2). The distribution of standard weighted practice scores approximated the normal distribution. About 76.89% mothers obtained a practice score ranging between -1 SD and +1 SD (between a score of 39.2 and 67) and 93.97% between -2 SD and +2 SD (Figs 4 and 5). The respondents scored lowest for physical distancing practice (mean 11.7, SD 9.8) followed by hand-washing and mask-wearing. The unweighted knowledge and practice scores demonstrated similar characteristics (Figs 2 and 4).

### Factors affecting COVID-19 preventive knowledge and practice

The log-odds of correct responses and the corresponding estimates of the association between the mothers' background information and COVID-19 preventive knowledge and practice are presented in Table 3. We found that mothers' educational qualification was significantly associated with the increased odds of correct response for COVID-19-related knowledge. Mothers with primary, secondary, and higher secondary education or above had 38% (CI: 1.24–1.54,

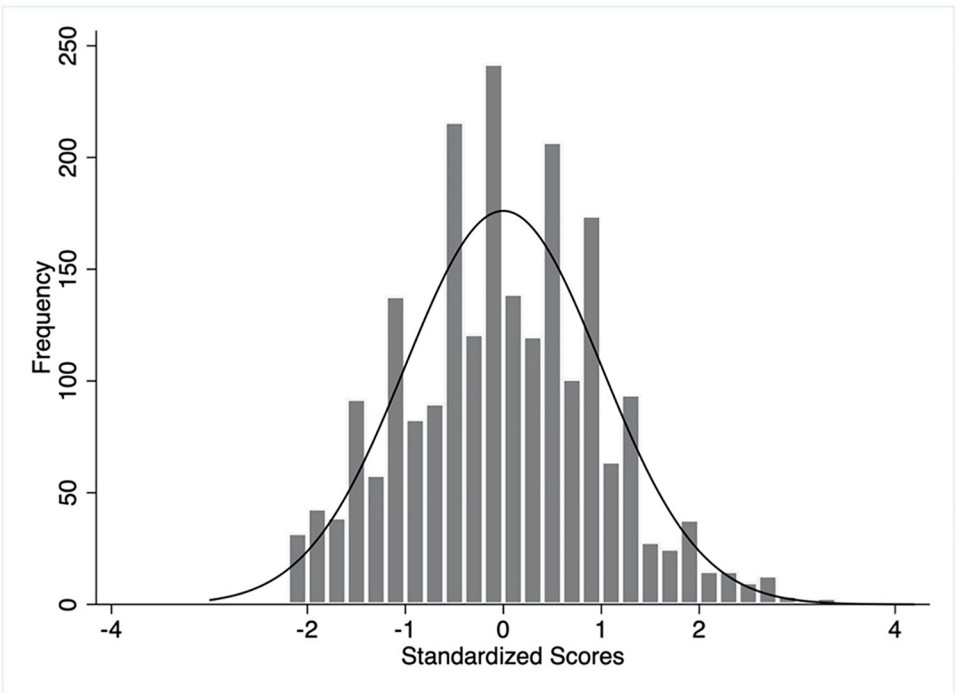

**Fig 2. Distributions of unweighted knowledge scores.**

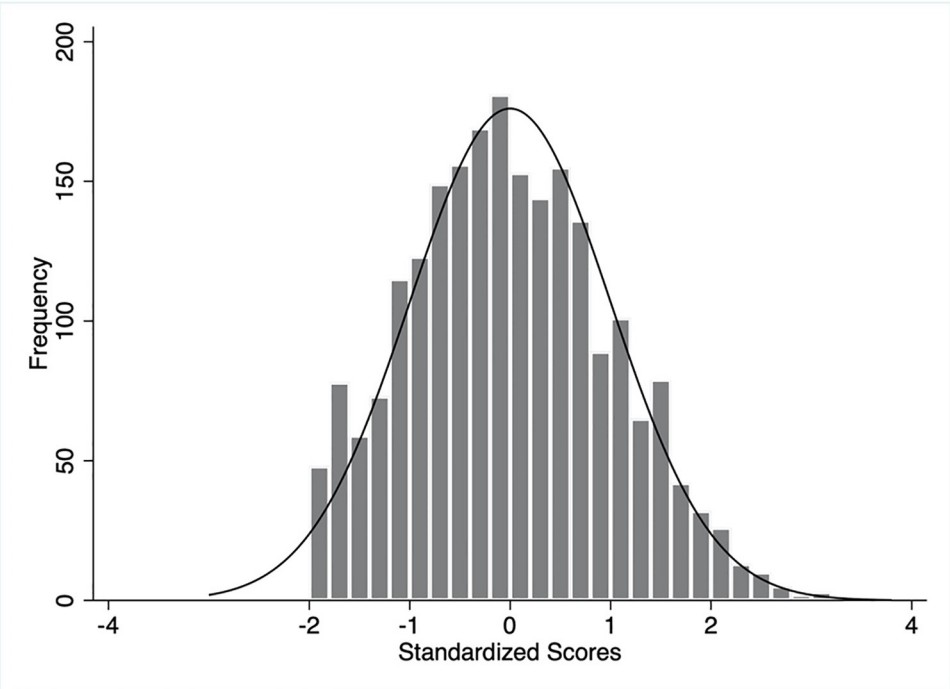

**Fig 3. Distributions of weighted knowledge scores.**

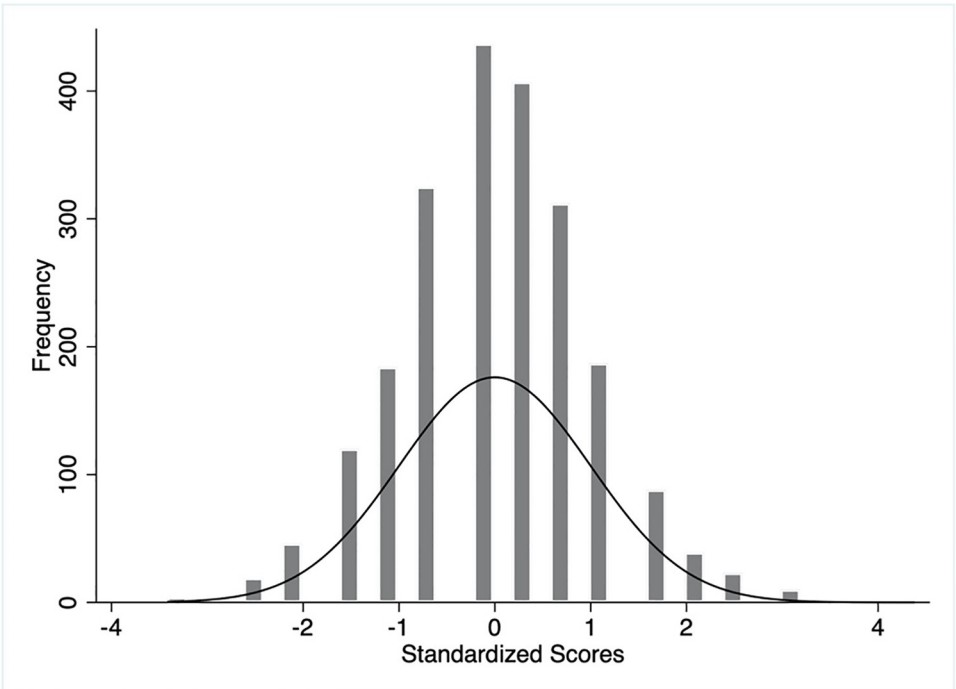

**Fig 4. Distributions of unweighted practice scores.**

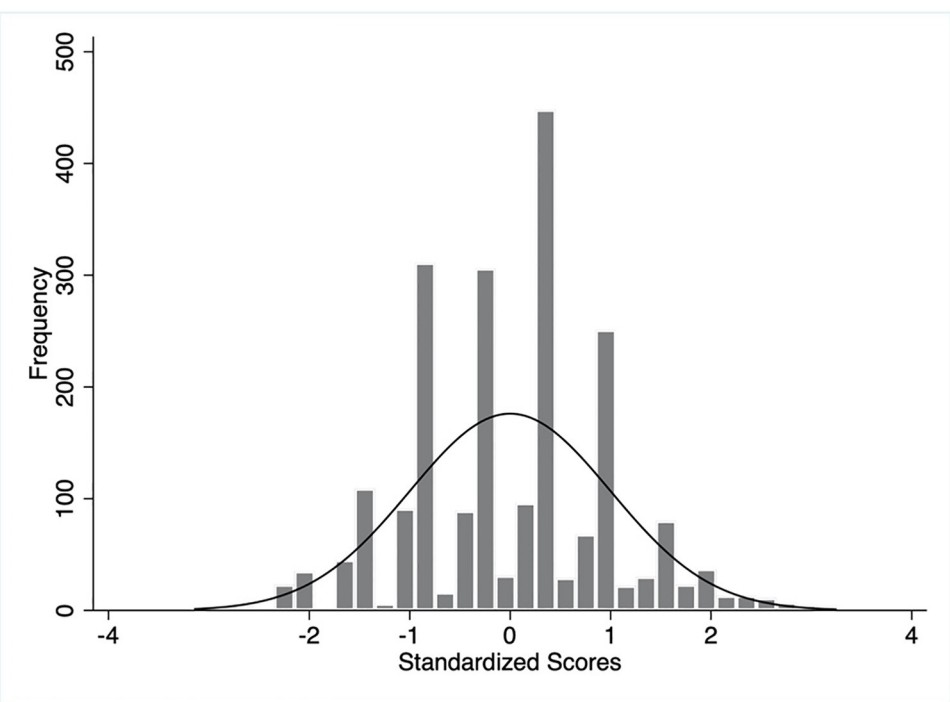

**Fig 5. Distributions of weighted practice scores.**

**Table 2. Unweighted and weighted scores.**

| | [1] | [2] | [3] | [2] | [3] |
|---|---|---|---|---|---|
| | **No. of Questions** | **Unweighted** | | **Weighted** | |
| | | **Scale** | **Score** | **Scale** | **Score** |
| **Knowledge related to COVID-19** | | | | | |
| Transmission, symptoms, and management | 4 | 0–22 | 6.8 (3.2) | 0–50 | 16.4 (8.3) |
| Prevention | 4 | 0–27 | 8.3 (4.5) | 0–50 | 16.2 (10.1) |
| Overall knowledge score | 8 | 0–49 | 15.1 (7.1) | 0–100 | 32.6 (16.4) |
| **Practices related to COVID-19** | | | | | |
| Handwashing | 4 | 0–14 | 5.9 (1.7) | 0–33.3 | 17.2 (4.3) |
| Mask-wearing | 4 | 0–4 | 2.9 (0.9) | 0–33.3 | 24.1 (7.3) |
| Physical distancing | 2 | 0–2 | 0.7 (0.6) | 0–33.3 | 11.7 (9.8) |
| Overall practice score | 10 | 0–20 | 9.5 (2.2) | 0–100 | 53.1 (13.9) |

**Note:** (a) Unless specified otherwise, mean knowledge- and practice-scores are reported; (b) standard deviation of the scores are reported in parentheses.

$p < .001$), 58% (CI: 1.41–1.76, $p < .001$), and 111% (CI: 1.84–2.41, $p < .001$) higher odds, respectively, of providing correct answers to knowledge questions related to COVID-19 prevention compared to mothers with no formal education. Likewise, the household heads' educational status had significant association with the mothers' knowledge level. On the other hand, service-holder household heads had 13% (CI: 1.03–1.23, $p = .008$) higher odds of delivering correct answers for COVID-19 knowledge in comparison to the wage-worker household heads. Mothers' knowledge score was also predicted by their access to television and internet. Those who had access to television and internet revealed 19% (CI: 1.13–1.26, $p < .001$) and 8% (CI: 1.02–1.14, $p = .006$) higher odds, respectively, of giving correct answers to knowledge questions compared to those who did not have access. Respondents from Bogura, Bagerhat, and Sherpur had 25% (CI: 1.02–1.52, $p = .030$), 38% (CI: 1.10–1.75, $p = .006$), 53% (CI: 1.19–1.94, $p = .001$) higher odds of offering accurate answers, respectively, on COVID-19 knowledge compared to Bhola. We found no significant association of mothers' knowledge with their age, occupation, household size, household income, and mobile phone access. In addition, the odds of the right answer regarding the knowledge of COVID-19 preventive measures was 4% higher than the odds of correct answer on transmission, symptoms, and management, which was statistically significant (Table 3).

In contrast, for the practice domain, the mask-wearing practices had 276% (CI: 3.57–3.96, $p < .001$) higher odds compared to the handwashing practice. On the other hand, the physical distancing practice had 25% (CI: 0.70–0.80, $p < .001$) lower odds in comparison to handwashing, which were statistically significant. Mothers with a higher secondary education or above had a 17% higher (CI: 1.05–1.31, $p = .006$) odds of practicing preventive measures compared to mothers with no formal education. Similarly, mothers' preventive practices had 12% (CI: 1.02–1.23, $p = .022$) higher odds in the households headed by individuals with higher secondary education or above compared to households headed by individuals with no formal education. Mothers from inlands, such as, Bogura, Bagerhat, and Sherpur had 16% (CI: 1.06–1.28, $p = .001$), 18% (CI: 1.06–1.31, $p = .002$), and 53% (CI: 1.11–1.38, $p < .001$) higher odds of following COVID-19 preventive precautions compared to wetlands or islands such as, Bhola. We found no statistically significant association between mothers' preventive practice score and their age, occupation, the occupation of the household head, household size, household income, or access to television and internet. Moreover, the mothers' COVID-19 preventive

**Table 3. Regression results.**

| | [1] | [2] |
|---|---|---|
| | **Knowledge** | **Practice** |
| **Knowledge domain** | | |
| Transmission, symptoms, and management | **Base** | |
| Prevention | 1.04 (0.011) | |
| | (1.01–1.06) | |
| **Practice domain** | | |
| Handwashing | | **Base** |
| Mask-wearing | | 3.76 (<0.001) |
| | | (3.57–3.96) |
| Physical-distancing | | 0.75 (<0.001) |
| | | (0.70–0.80) |
| Age (in years) | 1.00 (0.166) | 1.00 (0.462) |
| | (0.99–1.00) | (0.99–1.00) |
| **Education** | | |
| No formal education | **Base** | **Base** |
| Primary or less | 1.38 (<0.001) | 1.09 (0.048) |
| | (1.24–1.54) | (1.00–1.88) |
| Above primary up to SSC | 1.58 (<0.001) | 1.09 (0.059) |
| | (1.41–1.76) | (1.00–1.88) |
| HSC and above | 2.11 (<0.001) | 1.17 (0.006) |
| | (1.84–2.41) | (1.05–1.31) |
| **Occupation** | | |
| Homemaker | **Base** | **Base** |
| Work outside | 1.06 (0.317) | 0.95 (0.291) |
| | (0.94–1.20) | (0.86–1.05) |
| **Head's education** | | |
| No formal education | **Base** | **Base** |
| Primary or less | 1.11 (0.002) | 1.04 (0.119) |
| | (1.04–1.18) | (0.99–1.10) |
| Above primary up to SSC | 1.21 (<0.001) | 1.06 (0.059) |
| | (1.13–1.30) | (1.00–1.12) |
| HSC and above | 1.19 (0.003) | 1.12 (0.022) |
| | (1.06–1.34) | (1.02–1.23) |
| **Head's occupation** | | |
| Wage-worker | **Base** | **Base** |
| Agriculture | 0.99 (0.737) | 1.01 (0.855) |
| | (0.93–1.06) | (0.95–1.06) |
| Service-holder | 1.13 (0.008) | 1.05 (0.20) |
| | (1.03–1.23) | (0.98–1.13) |
| Self-employed | 1.07 (0.061) | 1.06 (0.069) |
| | (1.00–1.15) | (1.00–1.12) |
| Home-maker | 0.90 (0.294) | 0.99 (0.881) |
| | (0.74–1.10) | (0.84–1.16) |
| Others | 1.02 (0.825) | 1.07 (0.259) |
| | (0.89–1.16) | (0.95–1.19) |
| Household size (no. of members) | 1.00 (0.851) | 1.01 (0.182) |
| | (0.98–1.01) | (1.00–1.02) |

(*Continued*)

**Table 3.** (Continued)

|  | [1] | [2] |
|---|---|---|
|  | **Knowledge** | **Practice** |
| Household income (standardized) | 1.00 (0.382) | 1.00 (0.554) |
|  | (0.99–1.00) | (0.99–1.00) |
| = 1 if has access to television | 1.19 (<0.001) | 1.01 (0.575) |
|  | (1.13–1.26) | (0.98–1.06) |
| = 1 if has access to mobile phone | 0.97 (0.771) | 1.15 (0.142) |
|  | (0.78–1.21) | (0.96–1.37) |
| = 1 if has access to internet | 1.08 (0.006) | 0.99 (0.609) |
|  | (1.02–1.14) | (0.95–1.03) |
| = 1 if shares kitchen with others | 0.97 (0.357) | 1.03 (0.275) |
|  | (0.90–1.04) | (0.97–1.10) |
| = 1 if shares toilet with others | 0.92 (0.012) | 0.96 (0.076) |
|  | (0.87–0.98) | (0.91–1.00) |
| **District** |  |  |
| Bhola | **Base** | **Base** |
| Narayanganj | 1.15 (0.225) | 1.12 (0.032) |
|  | (0.92–1.45) | (1.01–1.25) |
| Bagerhat | 1.38 (0.006) | 1.18 (0.002) |
|  | (1.10–1.75) | (1.06–1.31) |
| Bogura | 1.25 (0.030) | 1.16 (0.001) |
|  | (1.02–1.52) | (1.06–1.28) |
| Kishoreganj | 0.81 (0.055) | 0.99 (0.754) |
|  | (0.66–1.00) | (0.90–1.08) |
| Sherpur | 1.53 (0.001) | 1.24 (<0.001) |
|  | (1.19–1.94) | (1.11–1.38) |
| Knowledge score (standardized) |  | 1.26 (<0.001) |
|  |  | (1.23–1.29) |
| Observations | 1,08,143 | 44,100 |
| Wald statistics (p-value) | 552.7 (<0.001) | 3282.6 (<0.001) |

**Note:** (a) *P*-values and 95% confidence intervals are reported in parentheses.

(b) Odds ratio is reported without parenthesis.

practice was significantly associated with their knowledge score. The odds of COVID-19 preventive practice was 26% greater among the mothers with knowledge of COVID-19 (Table 3).

## Discussion

This study provides evidence on the knowledge and practice related to COVID-19 prevention among mothers of under-2 children and identifies a few factors associated with these knowledge and practices.

The most prominent finding that emerged from this study is the inadequate level of COVID-19 preventive knowledge and practices among the respondents. It revealed that mothers of children under 2 lacked a comprehensive understanding of COVID-19 transmission, symptoms, and preventative practices, including appropriate mask-wearing techniques, handwashing guidelines, and social distancing. Individuals infected with COVID-19 will continue to spread the virus unless there is at least a basic awareness of common symptoms. Many

respondents were unaware that masks should not be shared, leading a majority to share masks despite having sufficient masks available. These findings are consistent with similar studies conducted in Bangladesh, Pakistan, and Iran among different vulnerable population groups, including slum dwellers, pregnant women, and the elderly, where the overall responses for knowledge level ranged between 27–44% [23,30–32]. If the community is unaware of the specific and detailed instructions for preventing COVID-19, even the most significant efforts to control the pandemic could fail.

In this study, we also found the respondents following substandard social distancing measures, inadequate hand-washing and mask-wearing practices. These findings of low COVID-19 preventive practices are comparable to the studies in Ethiopia and Nigeria, where practice levels were about 28% and 41%, respectively [33,34]. However, the deficient knowledge and practice levels in the present study are contrary to some of the prior studies, which suggest satisfactory knowledge level ranged between 80–90% in China, Malaysia, and Bangladesh [35–37] and practice level 76–93% in India, Vietnam, and China [37–39]. Furthermore, it is somewhat surprising that no significant change was observed between the current endline study and the previous baseline study conducted in the same districts, where the intervention was implemented for six months [29].

The suboptimal level of knowledge of our participants can be explained by their relatively lower education level (equal to or below the secondary level) and limited access to television and the internet leading to a reduced exposure to information. It is obvious that inadequate and poor level of knowledge negatively affects preventive practices. In addition to an appropriate knowledge level, COVID-19 preventive practices also depend on general health literacy, perceived risk of infection, individual and social motivation, and strict implementation of government rules and regulations to follow the preventive measures [36–38,39]. These help explain why the respondents did not follow the preventive precautions in the current study.

The regression results revealed that mothers who possessed knowledge about the transmission, symptoms, and management of COVID-19 were more inclined to be familiar with preventive measures such as mask-wearing, hand-washing, and social distancing. Those who had a higher knowledge on COVID-19 were more likely to adhere to these preventative measures. A significant practical implication of this finding is the necessity to emphasize comprehensive COVID-19 awareness alongside strict adherence to preventive practices while designing interventions. These findings are consistent with prior research conducted in Bangladesh, Ghana, and South Korea [40–42]. Furthermore, in this study, educated mothers demonstrated higher successful in acquiring adequate knowledge about COVID-19 and adhering to preventive practices, aligning with earlier evidence [28,42]. These findings suggest that individuals with low educational attainment may be more susceptible to COVID-19. Additionally, mothers with formal employment and access to television and the internet exhibited greater knowledge about COVID-19 compared to those with low education and no access to information, consistent with previous research [42,43]. This could be associated to information disseminated and government regulations taken during the pandemic in formal workplace settings. Therefore, media outlets may play a crucial role in planning long-term strategies to control the pandemic.

In this study, we found that wetlands exhibited significantly poorer levels of COVID-19 preventive knowledge and practices compared to all other districts. This finding aligns closely with another nationwide survey in Bangladesh [21]. The study sites in the wetlands were riverine islands (i.e., Char) and swamps (Haor) regions, which are hard to reach. Additionally, both districts have lower literacy rates compared to the other districts in the study [44]. An important implication of this finding is that effective pandemic control may require tailored interventions for risk communication and message dissemination in geographically isolated, hard-to-reach regions with lower educational levels.

### Limitations

One limitation of this study is the potential for reporting bias, as participants' self-reported data on COVID-19 preventive practices were collected through face-to-face interviews. This data was not triangulated by observations or other data collection methods, which may lead to overstated responses and potentially higher estimates of COVID-19 preventive practices. Additionally, participants were asked to recall their COVID-19 preventive measures from the past two weeks, which could introduce unintentional recall bias. As this was a cross-sectional study, causal inferences cannot be made from the findings. Unless for comparable settings, the results are also not generalizable to other settings or populations.

## Conclusion

In this study, mothers of under-2 children demonstrated an overall lack of knowledge and practices related to COVID-19 prevention. There was a significant association between respondents' COVID-19-related knowledge and their level of education and access to information. Moreover, mothers' overall education and specific knowledge of COVID-19 prevention are also interlinked with their preventive practices. Limitations in comprehensive understanding of COVID-19 and its prevention resulted in a significant drop in their preventive practices. Hence, individuals with lower levels of education and poor knowledge of COVID-19 are more vulnerable to COVID-19.

This study reinforces the importance of targeted interventions to improve COVID-19-related knowledge and practices among mothers of under-2 children, particularly those with limited education and living in geographically isolated regions, such as, char and haor areas. Improving their knowledge could help enhance their adherence to preventive measures against COVID-19. It is also essential to widely communicate detailed information about the transmission, symptoms, management, and prevention of COVID-19 through innovative channels where access to popular media is limited. Governments and development partners can design comprehensive community-based health education programs to develop awareness on COVID-19. It may eventually help maximize the uptake of essential MNCH services and reduce maternal and neonatal mortality in Bangladesh during pandemics like COVID-19.

## Supporting information

**S1 Table. Survey questions on knowledge related to COVID-19.**
(DOCX)

**S2 Table. Survey questions on practices related to COVID-19.**
(DOCX)

**S1 Dataset. De-identified data.**
(XLSX)

## Acknowledgments

We convey our heartfelt gratitude to all the survey participants, local administrative offices of the study areas, enumerators, data collection supervisors, and other research staff for their contribution to this study.

## Author Contributions

**Conceptualization:** Muhammed Nazmul Islam, Malabika Sarker.

**Data curation:** Antara Roy.

**Formal analysis:** Zarin Tasnim, Muhammed Nazmul Islam.

**Methodology:** Muhammed Nazmul Islam, Malabika Sarker.

**Project administration:** Antara Roy.

**Supervision:** Malabika Sarker.

**Writing – original draft:** Zarin Tasnim, Muhammed Nazmul Islam.

**Writing – review & editing:** Zarin Tasnim, Muhammed Nazmul Islam, Antara Roy, Malabika Sarker.

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
