## [Decision Letter · Decision Letter 0]

30 Apr 2024

PGPH-D-24-00415

Factors associated with knowledge and practices of COVID-19 prevention among mothers of under-2 children in Bangladesh

Dear Dr. Tasnim,

Thank you for submitting your manuscript to PLOS Global Public Health. After careful consideration, we feel that it has merit but does not fully meet PLOS Global Public Health’s publication criteria as it currently stands. Therefore, we invite you to submit a revised version of the manuscript that addresses the points raised during the review process.

Please make changes as suggested by both reviewers and submit your revised manuscript by 29th May, 2024. If you will need more time than this to complete your revisions, please reply to this message or contact the journal office at globalpubhealth@plos.org. Please include the following items when submitting your revised manuscript:

We look forward to receiving your revised manuscript.

Kind regards,

Preeti Mahato, Ph.D.

Academic Editor

Journal Requirements:

Additional Editor Comments (if provided):

Reviewers' comments:

Reviewer's Responses to Questions

**Comments to the Author**

1. Does this manuscript meet PLOS Global Public Health’s publication criteria? Is the manuscript technically sound, and do the data support the conclusions? The manuscript must describe methodologically and ethically rigorous research with conclusions that are appropriately drawn based on the data presented.

Reviewer #1: Partly

Reviewer #2: Yes

2. Has the statistical analysis been performed appropriately and rigorously?

Reviewer #1: Yes

Reviewer #2: Yes

3. Have the authors made all data underlying the findings in their manuscript fully available (please refer to the Data Availability Statement at the start of the manuscript PDF file)?

Reviewer #1: Yes

Reviewer #2: No

4. Is the manuscript presented in an intelligible fashion and written in standard English?

Reviewer #1: Yes

Reviewer #2: Yes

5. Review Comments to the Author

Reviewer #1: The manuscript reflects methodologically sound and ethically rigorous research, with the conclusions drawn mostly aligning with the data presented.

The statistical analyses seem to have been performed appropriately and rigorously.

The authors have also made all data underlying the findings fully available, enabling open science practices such as replications to comparable contexts.

Besides, the manuscript has been formulated in an intelligible fashion and written in standard English.

Nevertheless, some constructive feedback is provided (as comments within the PDF), which are believed to further improve its methodological rigour.

Reviewer #2: Interesting research has been conducted on the topic of "Factors associated with knowledge and practices of COVID-19 prevention among mothers of under-2 children in Bangladesh." For further quality improvement, it would be much appreciated if the authors addressed the following concerns in their paper:

Abstract:

• The conclusion should be based on the assessment of the findings. The author stated that tailored educational and behavioral interventions for mothers from lower socio-economic strata and hard-to-reach geographic locations could... What was the basis for this statement since there are no such variables aligned with this statement?

Introduction/Background:

• In lines 45-46, use either ‘Introduction’ or ‘Background’ following the journal's instructions.

• How do the study objective and findings align with fear and stigma, since these words are mentioned twice throughout the paper under background and conclusion? Since no relation to findings or discussion, what was the basis?

Methods:

• What was the procedure of the systematic random sampling approach to select the mothers from each village?

• Was there a minimum sample available in each selected village? In a yes or no situation, how was the sampling strategy applied? Does it vary from village to village?

Results and Discussions:

• It is suggested to include a table as a main table with the frequency distribution of knowledge and practice questions. This may ensure a better flow to understand Table 2.

• Based on the knowledge score and practice, how did the authors determine poor knowledge and practice? When can it be called good or better knowledge based on the score? What is the cutoff point for it?

• Tables are not clearly understandable, particularly table 3. Tables should be more structured with all the required information.

Limitation:

• This study only considered mothers from villages. Since mothers from urban areas were not considered, there is a concern with the generalization of the findings. Thus, it can be a limitation.

Conclusion:

• Overall, the study recommendation needs to be aligned with the study objective. Therefore, it’s suggested to revise in line with the findings.

Other Comments:

• There are line and word spacing problems throughout the paper.

• There are a few spelling mistakes and writing errors throughout the writing.

Overall, the paper is well-written. It would be appreciated if the authors addressed the above-mentioned concerns for further improvement.

6. PLOS authors have the option to publish the peer review history of their article (what does this mean?). If published, this will include your full peer review and any attached files.

**Do you want your identity to be public for this peer review?** For information about this choice, including consent withdrawal, please see our Privacy Policy.

Reviewer #1: **Yes: **Animesh Talukder

Reviewer #2: No

---

## [Decision Letter · Decision Letter 1]

2 Aug 2024

Factors associated with knowledge and practices of COVID-19 prevention among mothers of under-2 children in Bangladesh

PGPH-D-24-00415R1

Dear Dr Tasnim,

We are pleased to inform you that your manuscript 'Factors associated with knowledge and practices of COVID-19 prevention among mothers of under-2 children in Bangladesh' has been provisionally accepted for publication in PLOS Global Public Health.

Best regards,

Preeti Mahato, Ph.D.

Academic Editor

Reviewer Comments (if any, and for reference):

Reviewer's Responses to Questions

**Comments to the Author**

1. If the authors have adequately addressed your comments raised in a previous round of review and you feel that this manuscript is now acceptable for publication, you may indicate that here to bypass the “Comments to the Author” section, enter your conflict of interest statement in the “Confidential to Editor” section, and submit your "Accept" recommendation.

Reviewer #1: All comments have been addressed

Reviewer #2: All comments have been addressed

2. Does this manuscript meet PLOS Global Public Health’s publication criteria? Is the manuscript technically sound, and do the data support the conclusions? The manuscript must describe methodologically and ethically rigorous research with conclusions that are appropriately drawn based on the data presented.

Reviewer #1: Yes

Reviewer #2: Yes

3. Has the statistical analysis been performed appropriately and rigorously?

Reviewer #1: Yes

Reviewer #2: Yes

4. Have the authors made all data underlying the findings in their manuscript fully available (please refer to the Data Availability Statement at the start of the manuscript PDF file)?

Reviewer #1: Yes

Reviewer #2: Yes

5. Is the manuscript presented in an intelligible fashion and written in standard English?

Reviewer #1: Yes

Reviewer #2: Yes

6. Review Comments to the Author

Reviewer #1: The authors have made all the necessary revision or clarification in light of my review feedback, so the revised manuscript can now be considered eligible for publication.

Reviewer #2: (No Response)

7. PLOS authors have the option to publish the peer review history of their article (what does this mean?). If published, this will include your full peer review and any attached files.

**Do you want your identity to be public for this peer review?** For information about this choice, including consent withdrawal, please see our Privacy Policy.

Reviewer #1: **Yes: **Animesh Talukder

Reviewer #2: No
